# Static and Dynamic Analysis of Linear Piezoelectric Structures Using Higher Order Shear Deformation Theories

Konstantinos I. Ntaflos [1], Konstantinos G. Beltsios [2,*] and Evangelos P. Hadjigeorgiou [1,*]

1   Department of Materials Science and Engineering, University of Ioannina, 45110 Ioannina, Greece
2   School of Chemical Engineering, National Technical University of Athens, Zografou, 15780 Athens, Greece
*   Correspondence: kgbelt@mail.ntua.gr (K.G.B.); ehadjig@uoi.gr (E.P.H.)

**Abstract:** This paper explores the effects of shear deformation on piezoelectric materials and structures that often serve as substrate layers of multilayer composite sensors and actuators. Based on higher-order shear elastic deformation and electric potential distribution theories, a general mathematical model is derived. Governing equations and the associated boundary conditions for a piezoelectric beam are developed using a generalized Hamilton's principle. The static and dynamic behavior of the piezoelectric structure is investigated. A bending problem in static analysis and a free vibration problem in dynamic analysis are solved. The obtained results are in very good agreement with the results of the exact two dimensional solution available in the literature.

**Keywords:** piezoelectric beams; higher-order shear deformation theories; shear effects; Hamilton's principle; functional layers for composite sensors and actuators

## 1. Introduction

Elastic beams and plates are common structural elements in various structures. Piezoelectric beams and plates in the field of composite materials are of particular interest for their functional role in layered-composite sensors and actuators [1,2]. The more accurate way to analyze the mechanical properties of elastic beams and plates and study their response to different mechanical loads is using the three-dimensional mathematical theory of elasticity. Because of the complexity of the three-dimensional theory, the calculation and evaluation of static and dynamic characteristics is frequently attempted via the application of various simplified theories, described collectively as "technical or engineering theories" [3]. Nevertheless, it is well known that the elementary beam theory (ETB) in bending problems underestimates deflections and overestimates natural frequencies since it disregards the transverse shear deformation effect. Timoshenko [4] was the first to include refined effects such as rotatory inertia and shear deformation in the beam theory. This theory is widely referred to as Timoshenko beam theory (TBT) or first order shear deformation theory (FSDT).

The limitations of elementary theory of beam (ETB) and first order shear deformation theory (FSDT) led to the development of higher order shear deformation theories. There are many higher-order shear deformation theories available in the literature for static and dynamic analysis of elastic beams [5–11]. Ambartsumian [12] developed a bending theory of anisotropic plates and shallow shells. Kruszewski [13] studied the effect of transverse shear and rotary inertia on the natural frequency of a uniform beam. Reddy [14] has developed the well-known third-order shear deformation theory for the nonlinear analysis of plates with moderate thickness. The trigonometric shear deformation theories are presented by Touratier [15], Vlazov and Leontiev [3] and Stein [16] for thick beams. Soldatos [17] has developed hyperbolic shear deformation theory for homogeneous monoclinic plates. Karama et al. [18] studied the mechanical behavior of laminated composite beams by the new multi-layered laminated composite structures model with transverse

shear stress continuity. Sayyad [19,20] has carried out a comparison of various linear shear deformation theories for the free vibration analysis of thick isotropic beams. Study of the literature [9,10,19,21,22] indicates that the research work dealing with bending analysis of thick elastic beams using higher-order shear deformation theories is still in its early stage. Furthermore, although various technical theories for piezoelectric beams can be found in the literature [1,2,23–26], no systematic derivation of higher-order theories for static and dynamic analysis of piezoelectric beams is available.

In the present study, a systematic derivation of a general mathematical model for static and dynamic analysis of piezoelectric beams is presented; the model is based on higher-order shear elastic deformation and electric potential distribution theories. Using a generalized Hamilton's Principle [25,27] suitable for piezoelectric materials, the full set of equations of motion as well as the associated boundary conditions are determined for bending problems. Using this model, bending deflections (transverse displacement, rotation and electric potential) and natural frequencies (flexural and thickness shear mode frequencies) of a simply supported piezoelectric beam are calculated and compared with the results obtained using a two-dimensional model available in the relevant literature [28]. The obtained results are of practical importance for the more accurate design of layered-composite piezoelectric sensors and actuators in engineering applications.

More specifically, the form of the unified displacement field and unified electric potential field, and the strain–displacement and stress–strain relations, are presented in Section 2. The generalized Hamilton's principle for the piezoelectric beam is described analytically and the full set of equations of motion and associated boundary conditions are determined for bending problems in Section 3. The bending problem of a simply supported piezoelectric beam under transverse loads is solved in Section 4, while the corresponding free flexural vibration problem is solved in Section 5. Various numerical results and related diagrams are presented and discussed in Section 6. Potential applications and future work of immediate interest is discussed briefly in Section 7.

## 2. Unified Theoretical Formulation for the Piezoelectric Beam

Considering a piezoelectric ceramic beam as shown in Figure 1, any boundary and transverse loading conditions might apply. The beam under consideration occupies the region given by the following:

$$0 \leq x \leq L, \quad -b/2 \leq y \leq b/2, \quad -h/2 \leq z \leq h/2,$$

where $x$, $y$ and $z$ are Cartesian co-ordinates, $L$ is the length, $h$ the thickness and $b$ the width of the piezoelectric beam. The piezoelectric beam is subjected to transverse load of intensity $q(x)$ per unit length of the beam.

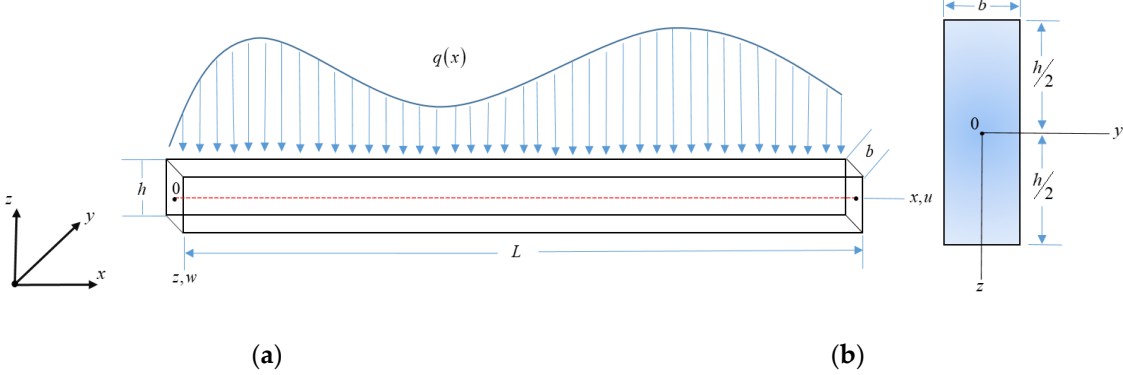

(**a**)  (**b**)

**Figure 1.** (**a**) Piezoelectric beam under bending in $x - z$ plane; (**b**) Cross-section of beam in $y - z$ plane.

### 2.1. The Displacement Field and Electric Potential

Based on the aforementioned assumptions, the displacement field [19] and the electric potential of the piezoelectric beam are given as below:

$$u(x,y,z,t) = -z\frac{\partial w(x,t)}{\partial x} + f(z)\varphi(x,t), \tag{1}$$

$$v(x,y,z,t) = 0, \tag{2}$$

$$w(x,y,z,t) = w(x,t), \tag{3}$$

$$\widetilde{\varphi}(x,y,z,t) = g(z)\overline{\varphi}(x,t). \tag{4}$$

Here, $u$ and $w$ are the axial and transverse displacements of the beam center line in $x$ and $z$ directions, respectively, $\widetilde{\varphi}$ is the electric potential and $t$ is the time. Symbol $\varphi$ represents the rotation of the cross-section of the beam at neutral axis, which is an unknown function to be determined. The functions $f(z)$, according to the shear stress distribution through the thickness of the beam, are given in Table 1.

**Table 1.** Function $f(z)$ for different high order shear stress distribution.

| Model | Function $f(z)$ |
|---|---|
| Ambartsumian (1958) [12] | $f(z) = \frac{z}{2}\left(\frac{h^2}{4} - \frac{z^2}{3}\right)$ |
| Kruszewski (1949) [13] | $f(z) = \frac{5z}{4}\left(1 - \frac{4z^2}{3h^2}\right)$ |
| Reddy (1990) [14] | $f(z) = z\left[1 - \frac{4}{3}\left(\frac{z}{h}\right)^2\right]$ |
| Touratier (1991) [15] | $f(z) = \frac{h}{\pi}sin\frac{\pi z}{h}$ |
| Soldatos (1992) [17] | $f(z) = z\,cosh\left(\frac{1}{2}\right) - h\,sinh\left(\frac{z}{h}\right)$ |
| Karama et al. (2003) [18] | $f(z) = z\,exp\left[-2\left(\frac{z}{h}\right)^2\right]$ |

The functions $g(z)$ describe the distribution of the electric potential inside the piezoelectric beam. According to the literature [18,21,23,24,26,29], various forms for $g(z)$ are given in Table 2. The form of the function $g(z)$ depends on the type of circuit conditions chosen for various applications. TYPE I refers to open circuit conditions and TYPE II refers to short circuit conditions.

**Table 2.** Function $g(z)$ describes the distribution of the electric potential along the thickness direction of the piezoelectric beam.

| Circuit Conditions | Function $g(z)$ | Author |
|---|---|---|
| TYPE I | $g(z) = -\frac{z}{h}$ | Goldschmidtboeing and Woias (2008) [21] |
| TYPE I | $g(z) = \frac{2z}{h}$ | Komeili et al. (2011) [30] |
| TYPE II | $g(z) = \frac{h}{\pi}cos\left(\frac{\pi z}{h}\right)$ | Fernandes and Pouget (2001) [29] |
| TYPE II | $g(z) = 1 - \left(\frac{2z}{h}\right)^2$ | Wang et al. (2001) [26] |
| TYPE II | $g(z) = cos\left(\frac{\pi z}{h}\right)$ | Baroudi et al. (2018) [23] |

The shear deformation effects are more important in thick beams than in slender beams. These effects are neglected in elementary bending theory of beams (Euler–Bernoulli Theory). To describe the correct bending behavior of thick beams including shear deformation effects, high order shear deformation theories are required. This can be achieved by the selection of proper kinematics and constitutive models.

The function $f(z)$ is included in the displacement field of high order theories to consider the effect of transverse shear deformation and achieve zero shear stress at the top and bottom surface of the beam [19].

### 2.2. Strain–Displacement and Stress–Strain Relations

Normal strain $\varepsilon_x$ and transverse shear strain $\gamma_{xz}$ for a linear piezoelectric beam are given by the following:

$$\varepsilon_x = \frac{\partial u}{\partial x} = -z\frac{\partial^2 w}{\partial x^2} + f(z)\frac{\partial \varphi}{\partial x}, \tag{5}$$

$$\gamma_{xz} = \frac{\partial u}{\partial z} + \frac{\partial w}{\partial x} = f'(z)\varphi(x,t). \tag{6}$$

Axial stress (normal bending stress) $\sigma_x$, transverse shear stress $\tau_{xz}$ and electric displacement components $D_x$ and $D_z$ for the piezoelectric beam [25,27] are given by the following:

$$\sigma_x = \widetilde{c}_{11}\varepsilon_x - \widetilde{e}_{31}E_z, \tag{7}$$

$$\tau_{xz} = c_{55}\gamma_{xz} - e_{15}E_x, \tag{8}$$

$$D_x = e_{15}\gamma_{xz} + \in_{11} E_x, \tag{9}$$

$$D_z = \widetilde{e}_{31}\varepsilon_x + \widetilde{\in}_{33}E_z, \tag{10}$$

where $\widetilde{c}_{11}$, $\widetilde{e}_{31}$ and $\widetilde{\in}_{33}$ are the reduced elastic, piezoelectric and dielectric constants under plane stress assumption from the 3-D constitutive relationship [26,31]; the latter constants are given by the following:

$$\begin{array}{lll}
\widetilde{c}_{11} = \overline{c}_{11} - \frac{\overline{c}_{13}^2}{\overline{c}_{33}} & , & \widetilde{e}_{31} = \overline{e}_{31} - \frac{\overline{c}_{13}\overline{e}_{33}}{\overline{c}_{33}} & , & \widetilde{\in}_{33} = \overline{\in}_{33} + \frac{\overline{e}_{33}^2}{\overline{c}_{33}} \\
\overline{c}_{11} = c_{11} - \frac{c_{12}^2}{c_{22}} & , & \overline{c}_{13} = c_{13} - \frac{c_{12}c_{23}}{c_{22}} & , & \overline{c}_{33} = c_{33} - \frac{c_{23}^2}{c_{22}} \\
\overline{e}_{31} = e_{31} - \frac{c_{12}e_{32}}{c_{22}} & , & \overline{e}_{33} = e_{33} - \frac{c_{23}e_{32}}{c_{22}} & , & \overline{\in}_{33} = \in_{33} + \frac{e_{32}^2}{c_{22}}.
\end{array} \tag{11}$$

Generally, the electric field vector for the piezoelectric beam in the quasi-electrostatic approximation is defined as follows:

$$E_i = -\frac{\partial \widetilde{\varphi}}{\partial x_i}. \tag{12}$$

Thus, using Equation (4), the components of the electric field in the beam across $x$, $y$ and $z$ directions are given by the following:

$$E_x = -\frac{\partial \widetilde{\varphi}}{\partial x} = -g(z)\frac{\partial \overline{\varphi}}{\partial x},$$

$$E_y = -\frac{\partial \widetilde{\varphi}}{\partial y} = 0, \tag{13}$$

$$E_z = -\frac{\partial \widetilde{\varphi}}{\partial z} = \frac{\partial g(z)}{\partial z}\overline{\varphi}(x,t) = -g'(z)\overline{\varphi}(x,t).$$

## 3. Governing Equations and Boundary Conditions

Using Equations (5) through (13) and Hamilton's principle, variationally consistent governing differential equations and boundary conditions for the piezoelectric beam under consideration can be obtained.

The generalized form of the Hamilton's principle for the piezoelectric beam (see also reference [25] for more details) is given as follows:

$$\delta \int_{t=t_1}^{t=t_2} (T - H_e + W_e)dt = 0,$$ (14)

where

$$\delta \int_{t=t_1}^{t=t_2} (T)dt = \delta \int_{t=t_1}^{t=t_2} \left[ \frac{1}{2} \int_V \rho \{\dot{u}\}^T \{\dot{u}\} dV \right] dt,$$ (15)

$$\delta \int_{t=t_1}^{t=t_2} (-H_e)dt = -\delta \int_{t=t_1}^{t=t_2} \int_V \left( \frac{1}{2} \{\varepsilon\}^T [C]\{\varepsilon\} - \{\varepsilon\}^T [e]^T \{E\} - \frac{1}{2} \{E\}^T [\in]\{E\} \right) dVdt,$$ (16)

$$\delta \int_{t=t_1}^{t=t_2} (W_e)dt = \delta \int_{t=t_1}^{t=t_2} \int_{x=0}^{x=L} qw(x,t)dxdt = \int_{t=t_1}^{t=t_2} \int_{x=0}^{x=L} q(\delta w)dxdt,$$ (17)

and $\delta(\cdot)$ denotes the first variation operator, $T$ is the kinetic energy, $H_e$ is the electric enthalpy and $W_e$ is the work performed by the external forces. $[C]$ is the elastic coefficient matric, $[e]$ is the piezoelectric coefficient matrix and $[\in]$ is the dielectric coefficient matrix. $\{u\}$ denotes the displacements field vector and $\{\dot{u}\}$ is the first time derivative of the displacements field vector. $\{\varepsilon\}$ is the strain vector, $\{E\}$ is the electric field vector and $q$ is the distributed forces along the length of the beam. Substituting Equations (15)–(17) into Equation (14), integrating by parts and collecting the coefficients of $\delta w$, $\delta \varphi$ and $\delta \overline{\varphi}$, the governing equations and the associated boundary conditions in terms of elastic displacements and electric field variables are derived.

The equations of motion for the piezoelectric beam are as follows:

$$-A_0 \tilde{c}_{11} b \frac{\partial^4 w}{\partial x^4} + B_0 \tilde{c}_{11} b \frac{\partial^3 \varphi}{\partial x^3} + A_0 \rho b \frac{\partial^4 w}{\partial x^2 \partial t^2} - B_0 \rho b \frac{\partial^3 \varphi}{\partial x \partial t^2} - \rho b h \frac{\partial^2 w}{\partial t^2} + \tilde{e}_{31} b K_0 \frac{\partial^2 \overline{\varphi}}{\partial x^2} + q = 0,$$ (18)

$$-B_0 \tilde{c}_{11} b \frac{\partial^3 w}{\partial x^3} + C_0 \tilde{c}_{11} b \frac{\partial^2 \varphi}{\partial x^2} - D_0 c_{55} b \varphi - C_0 \rho b \frac{\partial^2 \varphi}{\partial t^2} + B_0 \rho b \frac{\partial^3 w}{\partial x \partial t^2} + \tilde{e}_{31} b L_0 \frac{\partial \overline{\varphi}}{\partial x} - e_{15} b N_0 \frac{\partial \overline{\varphi}}{\partial x} = 0,$$ (19)

$$\tilde{e}_{31} b K_0 \frac{\partial^2 w}{\partial x^2} + e_{15} b N_0 \frac{\partial \varphi}{\partial x} - \tilde{e}_{31} b L_0 \frac{\partial \varphi}{\partial x} - P_0 b \in_{11} \frac{\partial^2 \overline{\varphi}}{\partial x^2} + M_0 b \tilde{\in}_{33} \overline{\varphi} = 0.$$ (20)

The boundary conditions at $x = 0$ and $x = L$ are as follows:

$$A_0 \tilde{c}_{11} b \frac{\partial^3 w}{\partial x^3} - B_0 \tilde{c}_{11} b \frac{\partial^2 \varphi}{\partial x^2} - A_0 \frac{\partial^3 w}{\partial x \partial t^2} + B_0 \frac{\partial^2 \varphi}{\partial t^2} - \tilde{e}_{31} b K_0 \frac{\partial \overline{\varphi}}{\partial x} = 0 \quad or \quad w \quad prescribed,$$ (21)

$$-A_0 \tilde{c}_{11} b \frac{\partial^2 w}{\partial x^2} + B_0 \tilde{c}_{11} b \frac{\partial \varphi}{\partial x} + \tilde{e}_{31} b K_0 \overline{\varphi}(x,t) = 0 \qquad or \quad \frac{\partial w}{\partial x} \quad prescribed,$$ (22)

$$B_0 \tilde{c}_{11} b \frac{\partial^2 w}{\partial x^2} - C_0 \tilde{c}_{11} b \frac{\partial \varphi}{\partial x} - \tilde{e}_{31} b L_0 \overline{\varphi}(x,t) = 0 \qquad or \quad \varphi \quad prescribed,$$ (23)

$$P_0 b \in_{11} \frac{\partial \overline{\varphi}}{\partial x} - e_{15} b N_0 \varphi(x,t) = 0 \qquad or \quad \overline{\varphi} \quad prescribed,$$ (24)

where the coefficients $A_0$, $B_0$, $C_0$, $D_0$, $K_0$, $L_0$, $M_0$, $N_0$ and $P_0$ are defined as follows:

$$A_0 = \int_{z=-\frac{h}{2}}^{z=\frac{h}{2}} z^2 dz, \quad B_0 = \int_{z=-\frac{h}{2}}^{z=\frac{h}{2}} z f(z) dz, \quad C_0 = \int_{z=-\frac{h}{2}}^{z=\frac{h}{2}} [f(z)]^2 dz,$$

$$D_0 = \int_{z=-\frac{h}{2}}^{z=\frac{h}{2}} [f'(z)]^2 dz, \quad K_0 = \int_{z=-\frac{h}{2}}^{z=\frac{h}{2}} z g'(z) dz, \quad L_0 = \int_{z=-\frac{h}{2}}^{z=\frac{h}{2}} f(z) g'(z) dz,$$ (25)

$$M_0 = \int_{z=-\frac{h}{2}}^{z=\frac{h}{2}} [g'(z)]^2 dz, \quad N_0 = \int_{z=-\frac{h}{2}}^{z=\frac{h}{2}} g(z) f'(z) dz, \quad P_0 = \int_{z=-\frac{h}{2}}^{z=\frac{h}{2}} [g(z)]^2 dz.$$

The coefficients $A_0$, $B_0$, $C_0$ and $D_0$ are related to the elastic properties and the coefficients $K_0$, $L_0$, $M_0$, $N_0$ and $P_0$ are related to the piezoelectric properties of the beam.

## 4. Bending Analysis of the Piezoelectric Beam

The bending problem of a simply supported piezoelectric beam which is subjected to transverse load $q(x)$, as shown in Figure 2, is solved. For a beam simply supported and grounded at two ends, the end conditions are given by $\sigma_x = w = \widetilde{\varphi} = 0$ at $x = 0$, $L$ and $-h/2 \leq z \leq h/2$.

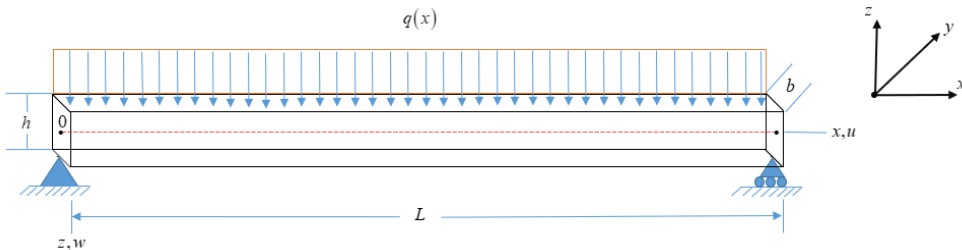

**Figure 2.** Geometry of a piezoelectric beam with simply supported boundary conditions, subjected to uniformly distributed load $q(x)$.

The governing equations for bending analysis of a piezoelectric beam (static case) are derived from Equations (18)–(20) discarding all terms containing time derivatives, as follows:

$$A_0\widetilde{c}_{11}b\frac{\partial^4 w}{\partial x^4} - B_0\widetilde{c}_{11}b\frac{\partial^3 \varphi}{\partial x^3} - K_0\widetilde{e}_{31}b\frac{\partial^2 \overline{\varphi}}{\partial x^2} = q, \tag{26}$$

$$B_0\widetilde{c}_{11}b\frac{\partial^3 w}{\partial x^3} - C_0\widetilde{c}_{11}b\frac{\partial^2 \varphi}{\partial x^2} + D_0 c_{55}b\varphi(x,t) - L_0\widetilde{e}_{31}b\frac{\partial \overline{\varphi}}{\partial x} + N_0 e_{15}b\frac{\partial \overline{\varphi}}{\partial x} = 0, \tag{27}$$

$$-K_0\widetilde{e}_{31}b\frac{\partial^2 w}{\partial x^2} + L_0\widetilde{e}_{31}b\frac{\partial \varphi}{\partial x} + P_0 b \in_{11} \frac{\partial^2 \overline{\varphi}}{\partial x^2} - N_0 e_{15}b\frac{\partial \varphi}{\partial x} - M_0 b\widetilde{\in}_{33}\overline{\varphi}(x,t) = 0. \tag{28}$$

The solution can be derived using the Fourier series method. The expansions are introduced in forms of sine and cosine series in order to satisfy the boundary conditions, as follows:

$$w(x) = \sum_{m=1}^{\infty} w_m \sin\frac{m\pi x}{L}, \tag{29}$$

$$\varphi(x) = \sum_{m=1}^{\infty} \varphi_m \cos\frac{m\pi x}{L}, \tag{30}$$

$$\overline{\varphi}(x) = \sum_{m=1}^{\infty} \overline{\varphi}_m \sin\frac{m\pi x}{L}, \tag{31}$$

$$q(x) = \sum_{m=1}^{\infty} q_m \sin\frac{m\pi x}{L}, \text{ with } q_m = \frac{4q_0}{m\pi} \text{ for } m = 1, 3, 5, \ldots$$
$$\text{and } q_m = 0 \quad \text{for } m = 2, 4, 6, \ldots \tag{32}$$

The expansions (Equations (29)–(32)) are introduced in Equations (26)–(28), and an algebraic system with tree variables results as follows:

$$\left(A_0\widetilde{c}_{11}b\frac{m^4\pi^4}{L^4}\right)w_m - \left(B_0\widetilde{c}_{11}b\frac{m^3\pi^3}{L^3}\right)\varphi_m + \left(\widetilde{e}_{31}bK_0\frac{m^2\pi^2}{L^2}\right)\overline{\varphi}_m = q_m, \tag{33}$$

$$-\left(B_0\widetilde{c}_{11}b\left(\frac{m\pi}{L}\right)^3\right)w_m + \left(C_0\widetilde{c}_{11}b\left(\frac{m\pi}{L}\right)^2 + D_0 c_{55}b\right)\varphi_m + (-L_0\widetilde{e}_{31}b + N_0 e_{15}b)\left(\frac{m\pi}{L}\right)\overline{\varphi}_m = 0, \tag{34}$$

$$K_0\widetilde{e}_{31}b\left(\frac{m\pi}{L}\right)^2 w_m + \left(-L_0\widetilde{e}_{31}b\left(\frac{m\pi}{L}\right) + N_0 e_{15}b\left(\frac{m\pi}{L}\right)\right)\varphi_m - \left(P_0 b \in_{11}\left(\frac{m\pi}{L}\right)^2 + M_0 b\widetilde{\in}_{33}\right)\overline{\varphi}_m = 0. \tag{35}$$

The above system is solved using the method of determinants, where the related determinants of the coefficients are as follows:

$$D = \det \begin{bmatrix} A_0\widetilde{c}_{11}b\frac{m^4\pi^4}{L^4} & -B_0\widetilde{c}_{11}b\frac{m^3\pi^3}{L^3} & K_0\widetilde{e}_{31}b\frac{m^2\pi^2}{L^2} \\ -B_0\widetilde{c}_{11}b\frac{m^3\pi^3}{L^3} & C_0\widetilde{c}_{11}b\frac{m^2\pi^2}{L^2} + D_0c_{55}b & -L_0\widetilde{e}_{31}b\frac{m\pi}{L} + N_0e_{15}b\frac{m\pi}{L} \\ K_0\widetilde{e}_{31}b\frac{m^2\pi^2}{L^2} & -L_0\widetilde{e}_{31}b\frac{m\pi}{L} + N_0e_{15}b\frac{m\pi}{L} & -P_0b \in_{11}\frac{m^2\pi^2}{L^2} - M_0b\widetilde{\in}_{33} \end{bmatrix}, \quad (36)$$

$$D_{w_m} = \det \begin{bmatrix} q_m & -B_0\widetilde{c}_{11}b\frac{m^3\pi^3}{L^3} & K_0\widetilde{e}_{31}b\frac{m^2\pi^2}{L^2} \\ 0 & C_0\widetilde{c}_{11}b\frac{m^2\pi^2}{L^2} + D_0c_{55}b & -L_0\widetilde{e}_{31}b\frac{m\pi}{L} + N_0e_{15}b\frac{m\pi}{L} \\ 0 & -L_0\widetilde{e}_{31}b\frac{m\pi}{L} + N_0e_{15}b\frac{m\pi}{L} & -P_0b \in_{11}\frac{m^2\pi^2}{L^2} - M_0b\widetilde{\in}_{33} \end{bmatrix}, \quad (37)$$

$$D_{\varphi_m} = \det \begin{bmatrix} A_0\widetilde{c}_{11}b\frac{m^4\pi^4}{L^4} & q_m & K_0\widetilde{e}_{31}b\frac{m^2\pi^2}{L^2} \\ -B_0\widetilde{c}_{11}b\frac{m^3\pi^3}{L^3} & 0 & -L_0\widetilde{e}_{31}b\frac{m\pi}{L} + N_0e_{15}b\frac{m\pi}{L} \\ K_0\widetilde{e}_{31}b\frac{m^2\pi^2}{L^2} & 0 & -P_0b \in_{11}\frac{m^2\pi^2}{L^2} - M_0b\widetilde{\in}_{33} \end{bmatrix}, \quad (38)$$

$$D_{\overline{\varphi}_m} = \det \begin{bmatrix} A_0\widetilde{c}_{11}b\frac{m^4\pi^4}{L^4} & -B_0\widetilde{c}_{11}b\frac{m^3\pi^3}{L^3} & q_m \\ -B_0\widetilde{c}_{11}b\frac{m^3\pi^3}{L^3} & C_0\widetilde{c}_{11}b\frac{m^2\pi^2}{L^2} + D_0c_{55}b & 0 \\ K_0\widetilde{e}_{31}b\frac{m^2\pi^2}{L^2} & -L_0\widetilde{e}_{31}b\frac{m\pi}{L} + N_0e_{15}b\frac{m\pi}{L} & 0 \end{bmatrix}. \quad (39)$$

Then, the general solution of the system has the following form:

$$w_m = \frac{q_m\left[\left(C_0\widetilde{c}_{11}ba_m^2 + D_0c_{55}b\right)\left(-P_0b \in_{11} a_m^2 - M_0b\widetilde{\in}_{33}\right) - \left(-L_0\widetilde{e}_{31}ba_m + N_0e_{15}ba_m\right)^2\right]}{\left[\left(A_0\widetilde{c}_{11}ba_m^4\left(C_0\widetilde{c}_{11}ba_m^2 + D_0c_{55}b\right) - \left(B_0\widetilde{c}_{11}ba_m^3\right)^2\right)\left(-P_0b \in_{11} a_m^2 - M_0b\widetilde{\in}_{33}\right) + \atop \begin{array}{l} +\left(-B_0\widetilde{c}_{11}ba_m^3\left(-L_0\widetilde{e}_{31}ba_m + N_0e_{15}ba_m\right) - \left(C_0\widetilde{c}_{11}ba_m^2 + D_0c_{55}b\right)\left(K_0\widetilde{e}_{31}ba_m^2\right)\right)\left(K_0\widetilde{e}_{31}ba_m^2\right) + \\ +\left(\left(-B_0\widetilde{c}_{11}ba_m^3\right)\left(K_0\widetilde{e}_{31}ba_m^2\right) - A_0\widetilde{c}_{11}ba_m^4\left(-L_0\widetilde{e}_{31}ba_m + N_0e_{15}ba_m\right)\right)\left(-L_0\widetilde{e}_{31}ba_m + N_0e_{15}ba_m\right)\end{array}\right]}, \quad (40)$$

$$\varphi_m = \frac{-q_m\left[-B_0\widetilde{c}_{11}ba_m^3\left(-P_0b \in_{11} a_m^2 - M_0b\widetilde{\in}_{33}\right) - \left(-L_0\widetilde{e}_{31}ba_m + N_0e_{15}ba_m\right)\left(K_0\widetilde{e}_{31}ba_m^2\right)\right]}{\left[\left(A_0\widetilde{c}_{11}ba_m^4\left(C_0\widetilde{c}_{11}ba_m^2 + D_0c_{55}b\right) - \left(B_0\widetilde{c}_{11}ba_m^3\right)^2\right)\left(-P_0b \in_{11} a_m^2 - M_0b\widetilde{\in}_{33}\right) + \atop \begin{array}{l} +\left(-B_0\widetilde{c}_{11}ba_m^3\left(-L_0\widetilde{e}_{31}ba_m + N_0e_{15}ba_m\right) - \left(C_0\widetilde{c}_{11}ba_m^2 + D_0c_{55}b\right)\left(K_0\widetilde{e}_{31}ba_m^2\right)\right)\left(K_0\widetilde{e}_{31}ba_m^2\right) + \\ +\left(\left(-B_0\widetilde{c}_{11}ba_m^3\right)\left(K_0\widetilde{e}_{31}ba_m^2\right) - A_0\widetilde{c}_{11}ba_m^4\left(-L_0\widetilde{e}_{31}ba_m + N_0e_{15}ba_m\right)\right)\left(-L_0\widetilde{e}_{31}ba_m + N_0e_{15}ba_m\right)\end{array}\right]}, \quad (41)$$

$$\overline{\varphi}_m = \frac{q_m\left[-B_0\widetilde{c}_{11}ba_m^3\left(-L_0\widetilde{e}_{31}ba_m + N_0e_{15}ba_m\right) - \left(C_0\widetilde{c}_{11}ba_m^2 + D_0c_{55}b\right)\left(K_0\widetilde{e}_{31}ba_m^2\right)\right]}{\left[\left(A_0\widetilde{c}_{11}ba_m^4\left(C_0\widetilde{c}_{11}ba_m^2 + D_0c_{55}b\right) - \left(B_0\widetilde{c}_{11}ba_m^3\right)^2\right)\left(-P_0b \in_{11} a_m^2 - M_0b\widetilde{\in}_{33}\right) + \atop \begin{array}{l} +\left(-B_0\widetilde{c}_{11}ba_m^3\left(-L_0\widetilde{e}_{31}ba_m + N_0e_{15}ba_m\right) - \left(C_0\widetilde{c}_{11}ba_m^2 + D_0c_{55}b\right)\left(K_0\widetilde{e}_{31}ba_m^2\right)\right)\left(K_0\widetilde{e}_{31}ba_m^2\right) + \\ +\left(\left(-B_0\widetilde{c}_{11}ba_m^3\right)\left(K_0\widetilde{e}_{31}ba_m^2\right) - A_0\widetilde{c}_{11}ba_m^4\left(-L_0\widetilde{e}_{31}ba_m + N_0e_{15}ba_m\right)\right)\left(-L_0\widetilde{e}_{31}ba_m + N_0e_{15}ba_m\right)\end{array}\right]}. \quad (42)$$

Using the above values of the coefficients $w_m$, $\varphi_m$ and $\overline{\varphi}_m$ in Equations (29)–(31), the final expressions of $w$, $\varphi$ and $\overline{\varphi}$ are obtained. Then, using the final expressions of $w$, $\varphi$ and $\overline{\varphi}$, in Equations (1), (3), (4), (7)–(10), final expressions for axial displacement $u$, transverse displacement $w$, electric potential $\widetilde{\varphi}$, axial bending stress $\sigma_x$, transverse shear stress $\tau_{xz}$, axial electric displacement $D_x$ and transverse electric displacement $D_z$ are obtained, as follows:

Axial displacement:

$$u = \sum_{m=1}^{\infty}\left\{\left[-z\frac{m\pi}{L}w_m + f(z)\varphi_m\right]cos\frac{m\pi x}{L}\right\}. \quad (43)$$

Transverse displacement:

$$w = \sum_{m=1}^{\infty}\left\{[w_m]sin\frac{m\pi x}{L}\right\}. \quad (44)$$

Electric potential:

$$\widetilde{\varphi} = \sum_{m=1}^{\infty} \left\{ [g(z)\overline{\varphi}_m] sin\frac{m\pi x}{L} \right\}. \tag{45}$$

Axial bending stress:

$$\sigma_x = \sum_{m=1}^{\infty} \left\{ \left[ \widetilde{c}_{11} z \frac{m^2\pi^2}{L^2} w_m - \widetilde{c}_{11} f(z) \frac{m\pi}{L} \varphi_m + \widetilde{e}_{31} g'(z) \overline{\varphi}_m \right] sin\frac{m\pi x}{L} \right\}. \tag{46}$$

Transverse shear stress:

$$\tau_{xz} = \sum_{m=1}^{\infty} \left\{ \left[ c_{55} f'(z) \varphi_m + e_{15} g(z) \frac{m\pi}{L} \overline{\varphi}_m \right] cos\frac{m\pi x}{L} \right\}. \tag{47}$$

Axial electric displacement:

$$D_x = \sum_{m=1}^{\infty} \left\{ \left[ e_{15} f'(z) \varphi_m - \in_{11} g(z) \frac{m\pi}{L} \overline{\varphi}_m \right] cos\frac{m\pi x}{L} \right\}. \tag{48}$$

Transverse electric displacement:

$$D_z = \sum_{m=1}^{\infty} \left\{ \left[ \widetilde{e}_{31} z \frac{m^2\pi^2}{L^2} w_m - \widetilde{e}_{31} f(z) \frac{m\pi}{L} \varphi_m - \widetilde{\in}_{33} g'(z) \overline{\varphi}_m \right] sin\frac{m\pi x}{L} \right\}. \tag{49}$$

## 5. Free Flexural Vibration Analysis of the Piezoelectric Beam

The governing equations for the free flexural vibration problem of a simply supported piezoelectric beam can be obtained by setting the applied transverse load $q(x)$ equal to zero in Equations (18)–(20). A solution of the resulting governing equations, which satisfies the associated initial conditions, is of the following form:

$$w(x,t) = \sum_{m=1}^{\infty} w_m sin\left(\frac{m\pi x}{L}\right) sin(\omega_m t), \tag{50}$$

$$\varphi(x,t) = \sum_{m=1}^{\infty} \varphi_m cos\left(\frac{m\pi x}{L}\right) sin(\omega_m t), \tag{51}$$

$$\overline{\varphi}(x,t) = \sum_{m=1}^{\infty} \overline{\varphi}_m sin\left(\frac{m\pi x}{L}\right) sin(\omega_m t), \tag{52}$$

where $w_m$, $\varphi_m$ and $\overline{\varphi}_m$ are the amplitudes of transverse displacement, rotation and electric potential, respectively, and $\omega_m$ is the natural frequency of the $m^{th}$ mode of vibration. Substitution of these solution forms into the governing equations of free vibration of piezoelectric beam results in the following linear algebraic equation system:

$$\left[ \left( A_0 \widetilde{c}_{11} b\alpha_m^4 \right) w_m - \left( B_0 \widetilde{c}_{11} b\alpha_m^3 \right) \varphi_m + \left( K_0 \widetilde{e}_{31} b\alpha_m^2 \right) \widetilde{\varphi}_m \right] -$$
$$-\omega_m^2 \left[ \left( A_o \rho b\alpha_m^2 + \rho bh \right) w_m - \left( B_o \rho ba_m \right) \varphi_m \right] = 0, \tag{53}$$

$$\left[ \left( -B_0 \widetilde{c}_{11} b\alpha_m^3 \right) w_m + \left( C_0 \widetilde{c}_{11} b\alpha_m^2 + D_0 c_{55} b \right) \varphi_m + \left( -L_0 \widetilde{e}_{31} ba_m + N_0 e_{15} ba_m \right) \widetilde{\varphi}_m \right] -$$
$$-\omega_m^2 \left[ \left( -B_o \rho ba_m \right) w_m + \left( C_o \rho b \right) \varphi_m \right] = 0, \tag{54}$$

$$\left[ \left( K_0 \widetilde{e}_{31} ba_m^2 \right) w_m + \left( -L_0 \widetilde{e}_{31} ba_m + N_0 e_{15} ba_m \right) \varphi_m - \left( P_0 \in_{11} ba_m^2 + M_0 \widetilde{\in}_{33} b \right) \widetilde{\varphi}_m \right] = 0. \tag{55}$$

The Equations (53)–(55) can be written in the following matrix form:

$$\left( \begin{bmatrix} K_{11} & K_{12} & K_{13} \\ K_{21} & K_{22} & K_{23} \\ K_{31} & K_{32} & K_{33} \end{bmatrix} - \omega_m^2 \begin{bmatrix} M_{11} & M_{12} & 0 \\ M_{21} & M_{22} & 0 \\ 0 & 0 & 0 \end{bmatrix} \right) \left\{ \begin{array}{c} w_m \\ \varphi_m \\ \widetilde{\varphi}_m \end{array} \right\} = 0. \tag{56}$$

Equation (56) can be written in the following, more compact form:

$$\left([K] - \omega_m^2[M]\right)\{\Delta\} = 0 \tag{57}$$

where $\{\Delta\}^T = \{w_m, \varphi_m, \widetilde{\varphi}_m\}$. The $[K]$ and $[M]$ are symmetric matrices, so we have $K_{12} = K_{21}$, $K_{13} = K_{31}$, $K_{23} = K_{32}$ and $M_{12} = M_{21}$.

The elements of the coefficient matrix $[K]$ are given by the following:

$$
\begin{aligned}
K_{11} &= A_0\widetilde{c}_{11}b\alpha_m^4, \\
K_{12} &= K_{21} = -B_0\widetilde{c}_{11}b\alpha_m^3, \\
K_{13} &= K_{31} = K_0\widetilde{e}_{31}b\alpha_m^2, \\
K_{22} &= C_0\widetilde{c}_{11}b\alpha_m^2 + D_0c_{55}b, \\
K_{23} &= K_{32} = -L_0\widetilde{e}_{31}ba_m + N_0e_{15}ba_m, \\
K_{33} &= -P_0 \in_{11} ba_m^2 - M_0\widetilde{\in}_{33}b.
\end{aligned}
\tag{58}
$$

The elements of the coefficient matrix $[M]$ are given by the following:

$$
\begin{aligned}
M_{11} &= A_o\rho b\alpha_m^2 + \rho bh, \\
M_{12} &= M_{21} = -B_o\rho ba_m, \\
M_{22} &= C_o\rho b.
\end{aligned}
\tag{59}
$$

For nontrivial solutions of Equation (57), the necessary condition is expressed as follows:

$$\det\left([K] - \omega_m^2[M]\right) = 0. \tag{60}$$

The solution of the above equation yields the values of the eigen-frequencies $\omega_m$ for various modes of vibration of the piezoelectric beam. Expanding Equation (60) gives the following:

$$\det\left(\begin{bmatrix} K_{11} & K_{12} & K_{13} \\ K_{21} & K_{22} & K_{23} \\ K_{31} & K_{32} & K_{33} \end{bmatrix} - \begin{bmatrix} \omega_m^2 M_{11} & \omega_m^2 M_{12} & 0 \\ \omega_m^2 M_{21} & \omega_m^2 M_{22} & 0 \\ 0 & 0 & 0 \end{bmatrix}\right) = 0. \tag{61}$$

After the necessary calculations, the following fourth order algebraic equation is obtained:

$$
\begin{aligned}
&\omega_m^4(K_{33}M_{11}M_{22} - K_{33}M_{12}M_{12}) + \\
&+\omega_m^2\left(\begin{array}{l} -K_{11}K_{33}M_{22} - K_{22}K_{33}M_{11} + K_{23}K_{23}M_{11} + K_{12}K_{33}M_{12} + \\ +K_{12}K_{33}M_{12} - K_{13}K_{23}M_{12} - K_{13}K_{23}M_{12} + K_{13}K_{13}M_{22} \end{array}\right) + \\
&+(K_{11}K_{22}K_{33} - K_{11}K_{23}K_{23} - K_{12}K_{12}K_{33} + K_{12}K_{13}K_{23} + K_{12}K_{13}K_{23} - K_{13}K_{13}K_{22}) = 0.
\end{aligned}
\tag{62}
$$

From the solution of the above equation, four roots are obtained: two positive frequencies $\omega_w$, $\omega_\varphi$ and two negative conjugate frequencies which are rejected. The first frequency $\omega_w$ is the flexural frequency and the second $\omega_\varphi$ is the fundamental frequency of thickness shear mode of the piezoelectric beam. The results for the fundamental frequency $\omega_m$ are presented in the following non-dimensional form:

$$\overline{\omega} = \omega_m\left(\frac{L^2}{h}\right)\sqrt{\frac{\rho}{\widetilde{c}_{11}}}. \tag{63}$$

## 6. Numerical Results and Discussion

For the examples and numerical calculations, a simply supported piezoelectric beam is considered, as shown in Figure 2, with length $L = 0.6$ m, thickness $b = 0.002$ m and height $h$. The value of the height $h$ depends on the aspect ratio $(S = L/h)$ values. Zero electric potential $\widetilde{\varphi}$ at the upper and lower $(x, z = \pm h/2)$ surface of the beam is considered (short circuit electric conditions) and for this reason, a TYPE II expression of the function $g(z) = 1 - \left(\frac{2z}{h}\right)^2$ is chosen. In the static problem, the beam is subjected to a uniformly

distributed exterior load $q_0 = 10 \, \text{N/m}^2$ ($q^* = q_0/b$ for the results presented below). The piezoelectric material of the beam is PZT–4 [28], with density $\rho = 7500 \, \text{kg/m}^3$ and an elastic coefficient matrix as follows:

$$[C] = \begin{bmatrix} c_{11} & c_{12} & c_{13} & 0 & 0 & 0 \\ c_{21} & c_{22} & c_{23} & 0 & 0 & 0 \\ c_{31} & c_{32} & c_{33} & 0 & 0 & 0 \\ 0 & 0 & 0 & c_{44} & 0 & 0 \\ 0 & 0 & 0 & 0 & c_{55} & 0 \\ 0 & 0 & 0 & 0 & 0 & c_{66} \end{bmatrix} = \begin{bmatrix} 139 & 77.8 & 74.3 & 0 & 0 & 0 \\ 77.8 & 139 & 74.3 & 0 & 0 & 0 \\ 74.3 & 74.3 & 11.3 & 0 & 0 & 0 \\ 0 & 0 & 0 & 25.6 & 0 & 0 \\ 0 & 0 & 0 & 0 & 25.6 & 0 \\ 0 & 0 & 0 & 0 & 0 & 30.6 \end{bmatrix} \text{GPa.} \quad (64)$$

It has a piezoelectric coefficient matrix as follows:

$$[e] = \begin{bmatrix} 0 & 0 & 0 & 0 & e_{15} & 0 \\ 0 & 0 & 0 & e_{15} & 0 & 0 \\ e_{31} & e_{32} & e_{33} & 0 & 0 & 0 \end{bmatrix} = \begin{bmatrix} 0 & 0 & 0 & 0 & 13.44 & 0 \\ 0 & 0 & 0 & 13.44 & 0 & 0 \\ -6.98 & -6.98 & 13.84 & 0 & 0 & 0 \end{bmatrix} \frac{\text{C}}{\text{m}^2}. \quad (65)$$

Moreover, it has a dielectric coefficient matrix as follows:

$$[\in] = \begin{bmatrix} \in_{11} & 0 & 0 \\ 0 & \in_{22} & 0 \\ 0 & 0 & \in_{33} \end{bmatrix} = \begin{bmatrix} 6.00 & 0 & 0 \\ 0 & 6.00 & 0 \\ 0 & 0 & 5.47 \end{bmatrix} \times 10^{-9} \frac{\text{V}}{\text{C} * \text{m}}. \quad (66)$$

### 6.1. Bending Analysis of the Piezoelectric Beam

For the bending analysis of the piezoelectric beam, the transverse displacement $w$, the rotation $\varphi$ and the electric potential $\widetilde{\varphi}$ are calculated for all models of the beam with various aspect ratios ($S = 2, 5, 10, 30$), and the obtained results are presented in Table 3.

**Table 3.** Comparison of transverse displacement $w$ at the center of the beam ($x = L/2$), rotation $\varphi$ at the end of the beam ($x = 0$) and electric potential $\widetilde{\varphi}$ at the center of the beam ($x = L/2$, $z = 0$) for linear high order shear deformation theories of piezoelectric beams with aspect ratios $S = 2, 5, 10, 30$.

| Aspect Ratio $S$ | Model | Transverse Displacement $w(m)$ | Rotation $\varphi(^o)$ | Electric Potential $\widetilde{\varphi}(V)$ |
|---|---|---|---|---|
| 2 | Ambartsumian [12] | $-1.2801112 \times 10^{-10}$ | $2.0286439 \times 10^{-8}$ | $-0.039315504$ |
|  | Kruszewski [13] | $-1.2803252 \times 10^{-10}$ | $5.1231438 \times 10^{-8}$ | $-0.039319084$ |
|  | Reddy [14] | $-1.2803614 \times 10^{-10}$ | $2.2822244 \times 10^{-8}$ | $-0.039315504$ |
|  | Touratier [15] | $-1.2794482 \times 10^{-10}$ | $2.3494154 \times 10^{-8}$ | $-0.039296693$ |
|  | Karama [18] | $-1.2752185 \times 10^{-10}$ | $2.4067348 \times 10^{-8}$ | $-0.039176182$ |
|  | Soldatos [17] | $-1.2800025 \times 10^{-10}$ | $2.1782638 \times 10^{-8}$ | $-0.040582753$ |
|  | Exact Solution [28] | $-1.2955144 \times 10^{-10}$ | $-$ | $-$ |
| 5 | Ambartsumian [12] | $-1.3902144 \times 10^{-9}$ | $3.5580401 \times 10^{-8}$ | $-0.071643269$ |
|  | Kruszewski [13] | $-1.3906821 \times 10^{-9}$ | $3.1694925 \times 10^{-8}$ | $-0.071660602$ |
|  | Reddy [14] | $-1.3903140 \times 10^{-9}$ | $6.4044722 \times 10^{-8}$ | $-0.071643269$ |
|  | Touratier [15] | $-1.3905663 \times 10^{-9}$ | $6.6076112 \times 10^{-8}$ | $-0.071656142$ |
|  | Karama [18] | $-1.3896790 \times 10^{-9}$ | $6.7971740 \times 10^{-8}$ | $-0.071614474$ |
|  | Soldatos [17] | $-1.3902219 \times 10^{-9}$ | $5.0052762 \times 10^{-8}$ | $-0.072021637$ |
|  | Exact Solution [28] | $-1.3854365 \times 10^{-9}$ | $-$ | $-$ |
| 10 | Ambartsumian [12] | $-1.0365670 \times 10^{-8}$ | $1.2937340 \times 10^{-7}$ | $-0.134659590$ |
|  | Kruszewski [13] | $-1.0369451 \times 10^{-8}$ | $1.0574177 \times 10^{-7}$ | $-0.134695396$ |
|  | Reddy [14] | $-1.0366002 \times 10^{-8}$ | $1.3218032 \times 10^{-7}$ | $-0.134659590$ |
|  | Touratier [15] | $-1.0369253 \times 10^{-8}$ | $1.3643415 \times 10^{-7}$ | $-0.134693113$ |
|  | Karama [18] | $-1.0367532 \times 10^{-8}$ | $1.4046581 \times 10^{-7}$ | $-0.134672500$ |
|  | Soldatos [17] | $-1.0365697 \times 10^{-8}$ | $1.0331467 \times 10^{-7}$ | $-0.134839081$ |
|  | Exact Solution [28] | $-1.0353429 \times 10^{-8}$ | $-$ | $-$ |

**Table 3.** *Cont.*

| Aspect Ratio $S$ | Model | Transverse Displacement $w(m)$ | Rotation $\varphi(^o)$ | Electric Potential $\widetilde{\varphi}(V)$ |
|---|---|---|---|---|
| 30 | Ambartsumian [12] | $-2.7373816 \times 10^{-7}$ | $3.4080394 \times 10^{-7}$ | $-0.396536889$ |
| | Kruszewski [13] | $-2.7384038 \times 10^{-7}$ | $3.2157700 \times 10^{-7}$ | $-0.396645338$ |
| | Reddy [14] | $-2.7374235 \times 10^{-7}$ | $4.0197111 \times 10^{-7}$ | $-0.396536889$ |
| | Touratier [15] | $-2.7383981 \times 10^{-7}$ | $4.1498035 \times 10^{-7}$ | $-0.396644474$ |
| | Karama [18] | $-2.7383470 \times 10^{-7}$ | $4.2737985 \times 10^{-7}$ | $-0.396637526$ |
| | Soldatos [17] | $-2.7373825 \times 10^{-7}$ | $3.1420304 \times 10^{-7}$ | $-0.396597288$ |
| | Exact Solution [28] | $-2.7370209 \times 10^{-7}$ | – | $-0.397$ |

Comparison of transverse displacement $w$ and electric potential $\widetilde{\varphi}$ for all models with higher order shear deformation terms through the length of the beam for various aspect ratios ($S = 2, 5, 10, 30$) are presented in Figures 3 and 4, respectively.

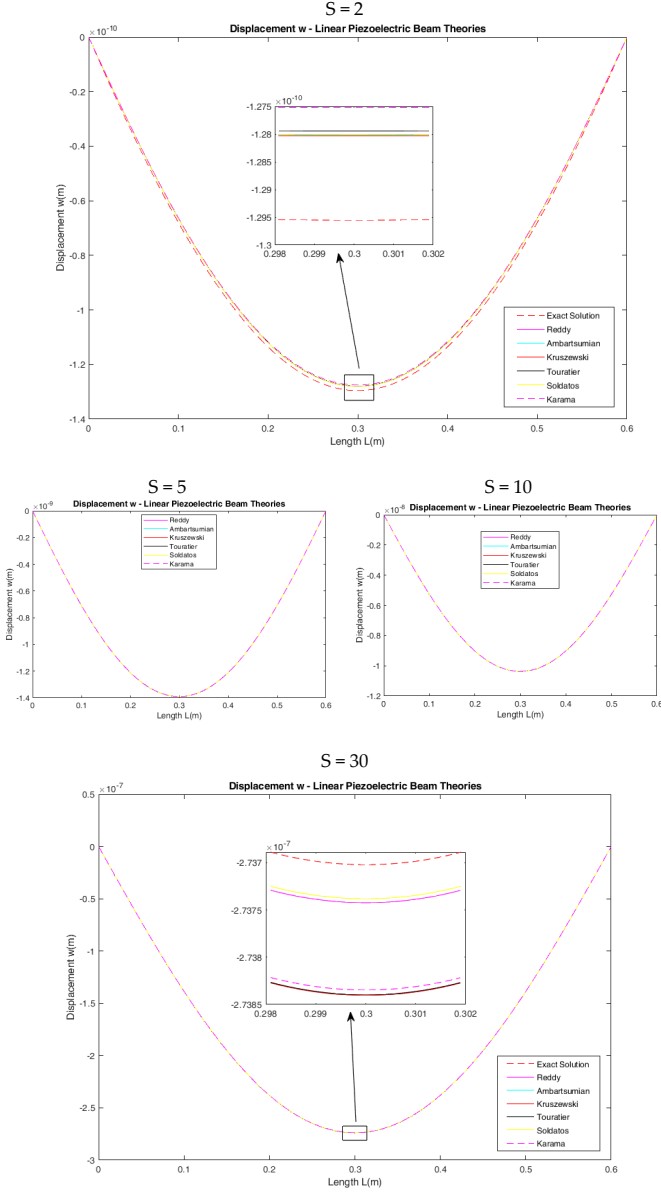

**Figure 3.** Comparison of transverse displacement $w$ through the length of the beam at $(x, z = 0)$ for various aspect ratios ($S = 2, 5, 10, 30$) and for all models with higher order shear deformation terms.

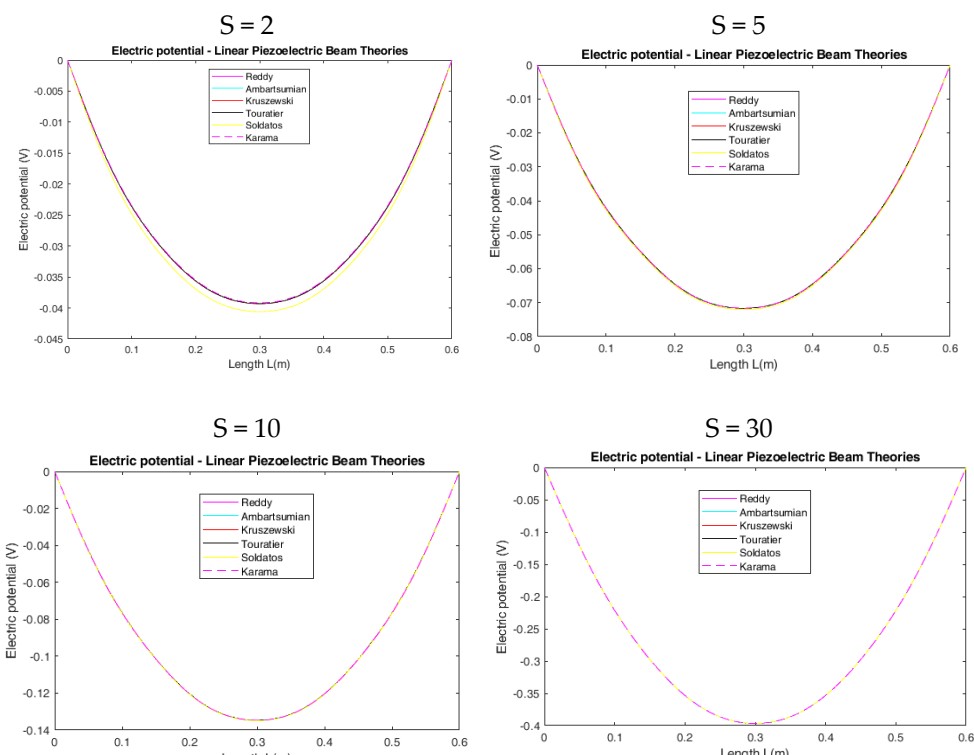

**Figure 4.** Comparison of electric potential $\widetilde{\varphi}$ through the length of the beam at $(x, z = 0)$ for various aspect ratios $(S = 2, 5, 10, 30)$ and for all models with higher order shear deformation terms.

From the above numerical results, it is concluded that the general model underestimates the maximum transverse displacement $w$ for very thick beams $(S = 2)$, while it overestimates $w$ for thick $(S = 5, S = 10)$ and slender $(S = 30)$ beams. The maximum electric potential $\widetilde{\varphi}$ predicted by the general model is in very good agreement with the exact solution for slender beams $(S = 30)$.

Comparison of axial stress $\sigma_x$ through the length of the beam at $(x, z = h/2)$ and through the height of the beam at $(x = L/2, z)$ for piezoelectric beams with aspect ratio $S = 30$ and for all models with higher order shear deformation terms are presented in Figures 5 and 6, respectively. In addition, comparison of transverse shear stress $\tau_{xz}$ through the length of the beam at $(x, z = h/4)$ and through the height of the beam at $(x = L/4, z)$ for all models with aspect ratio $S = 30$ are presented in Figures 7 and 8, respectively.

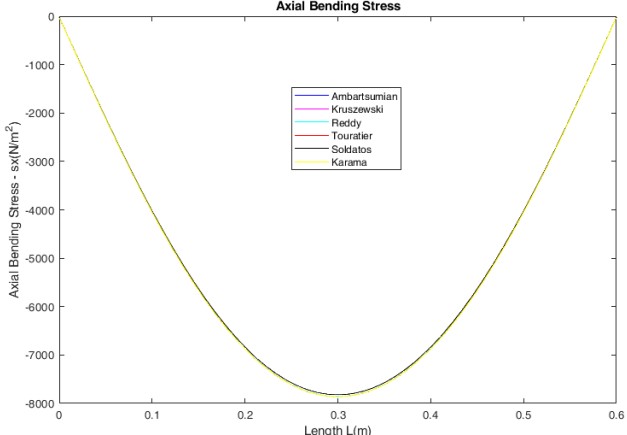

**Figure 5.** Comparison of axial stress $\sigma_x$ through the length of the beam at $(x, z = h/2)$ for all models with aspect ratio $S = 30$.

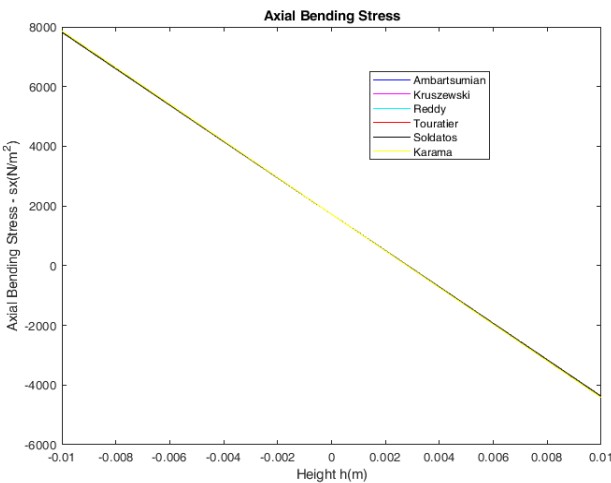

**Figure 6.** Comparison of axial stress $\sigma_x$ through the height of the beam at $(x = L/2,\ z)$ for all models with aspect ratio $S = 30$.

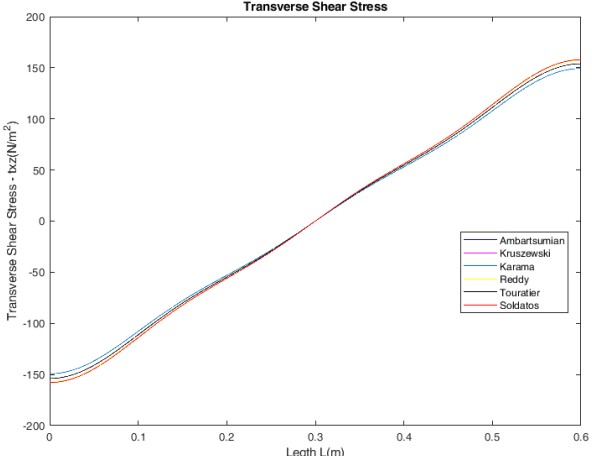

**Figure 7.** Comparison of transverse shear stress $\tau_{xz}$ through the length of the beam at $(x,\ z = h/4)$ for all models with aspect ratio $S = 30$.

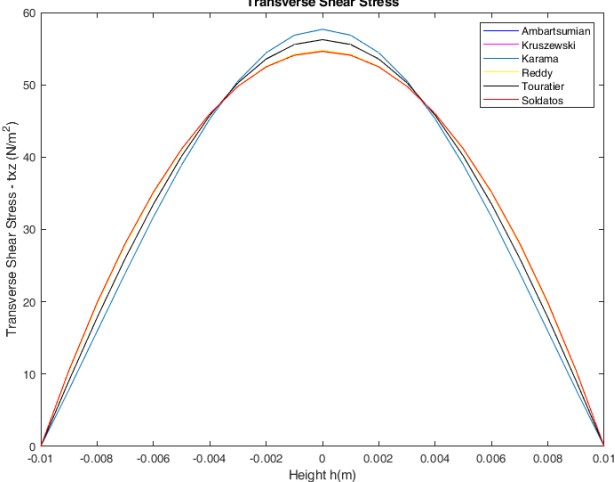

**Figure 8.** Comparison of transverse shear stress $\tau_{xz}$ through the height of the beam at $(x = L/4,\ z)$ for all models with aspect ratio $S = 30$.

From the numerical results presented in Table 4, it follows that the axial bending stress $\sigma_x$ predicted by all models is in very good agreement with the exact solution for beams with aspect ratio $S = 30$. The maximum transverse shear stress $\tau_{xz}$ predicted by the Touratier model is in very good agreement with that from the exact solution. The models of Ambartsumian, Reddy and Soldatos underestimate while the model of Karama overestimates the value of transverse shear stress for beams with aspect ratio $S = 30$.

**Table 4.** Axial stress $\sigma_x$ at $(x = L/2, z = -h/2)$ and transverse shear stress $\tau_{xz}$ at $(x = L/4, z = 0)$ for high order shear deformation models of piezoelectric beams with aspect ratio $S = 30$.

| Models | Ambartsumian [12] | Kruszewski [13] | Reddy [14] | Touratier [15] | Karama [18] | Soldatos [17] | Exact Solution [28] |
|---|---|---|---|---|---|---|---|
| $\sigma_x \left(\frac{N}{m^2}\right)$ | $-7858.228$ | $-7860.983$ | $-7858.228$ | $-7860.700$ | $-7860.203$ | $-7858.250$ | $-7860$ |
| $\tau_{xz} \left(\frac{N}{m^2}\right)$ | $54.7450$ | $54.7477$ | $54.7450$ | $56.2460$ | $57.6778$ | $54.6084$ | $56$ |

*6.2. Free Flexural Vibration Analysis of the Piezoelectric Beam*

For the free flexural vibration analysis of the piezoelectric beam, the non-dimensional flexural frequency $\overline{\omega}_w$ and the non-dimensional frequency of thickness shear mode $\overline{\omega}_\varphi$ is calculated for various modes of vibration and various aspect ratios. The results are presented in Tables 5–7.

**Table 5.** Comparison of non-dimensional fundamental ($m = 1$) flexural and thickness shear mode frequencies of the piezoelectric beam.

| Model | S=2 | | S=5 | | S=10 | | S=30 | |
|---|---|---|---|---|---|---|---|---|
| | $\overline{\omega}_w$ | $\overline{\omega}_\varphi$ | $\overline{\omega}_w$ | $\overline{\omega}_\varphi$ | $\overline{\omega}_w$ | $\overline{\omega}_\varphi$ | $\overline{\omega}_w$ | $\overline{\omega}_\varphi$ |
| Ambartsumian [12] | 2.3315 | 9.8072 | 2.8681 | 47.5546 | 3.0006 | 180.3669 | 3.0454 | 1595.544 |
| Kruszewski [13] | 2.3315 | 9.8072 | 2.8681 | 47.5546 | 3.0006 | 180.3669 | 3.0454 | 1595.544 |
| Reddy [14] | 2.3315 | 9.8072 | 2.8681 | 47.5546 | 3.0006 | 180.3669 | 3.0454 | 1595.544 |
| Touratier [15] | 2.3325 | 9.8022 | 2.8683 | 47.5255 | 3.0007 | 180.2522 | 3.0454 | 1594.517 |
| Karama [18] | 2.3358 | 9.8048 | 2.8692 | 47.5546 | 3.0010 | 180.3821 | 3.0454 | 1595.728 |
| Soldatos [17] | 2.3314 | 9.8080 | 2.8681 | 47.5599 | 3.0007 | 180.3882 | 3.0454 | 1595.736 |

**Table 6.** Comparison of non-dimensional flexural frequency $\overline{\omega}_w$ of the piezoelectric beam for various modes of vibration.

| Aspect Ratio S | Model | Modes of Vibration | | | | |
|---|---|---|---|---|---|---|
| | | $m=1$ | $m=2$ | $m=3$ | $m=4$ | $m=5$ |
| 2 | Ambartsumian [12] | 2.3315 | 6.7289 | 11.7170 | 16.9055 | 22.1801 |
| | Kruszewski [13] | 2.3315 | 6.7289 | 11.7170 | 16.9055 | 22.1801 |
| | Reddy [14] | 2.3315 | 6.7289 | 11.7170 | 16.9055 | 22.1801 |
| | Touratier [15] | 2.3325 | 6.7382 | 11.7464 | 16.9693 | 22.2945 |
| | Karama [18] | 2.3358 | 6.7614 | 11.8122 | 17.1035 | 22.5246 |
| | Soldatos [17] | 2.3314 | 6.7280 | 11.7141 | 16.8994 | 22.1693 |
| 5 | Ambartsumian [12] | 2.8681 | 10.0322 | 19.5357 | 30.3754 | 42.0556 |
| | Kruszewski [13] | 2.8681 | 10.0322 | 19.5357 | 30.3754 | 42.0556 |
| | Reddy [14] | 2.8681 | 10.0322 | 19.5357 | 30.3754 | 42.0556 |
| | Touratier [15] | 2.8683 | 10.0350 | 19.5474 | 30.4050 | 42.1138 |
| | Karama [18] | 2.8692 | 10.0451 | 19.5826 | 30.4846 | 42.2589 |
| | Soldatos [17] | 2.8681 | 10.0319 | 19.5346 | 30.3726 | 42.0500 |

**Table 6.** *Cont.*

| Aspect Ratio S | Model | Modes of Vibration | | | | |
|---|---|---|---|---|---|---|
| | | *m*=1 | *m*=2 | *m*=3 | *m*=4 | *m*=5 |
| 10 | Ambartsumian [12] | 3.0006 | 11.4726 | 24.2367 | 40.1288 | 58.2890 |
| | Kruszewski [13] | 3.0006 | 11.4726 | 24.2367 | 40.1288 | 58.2890 |
| | Reddy [14] | 3.0006 | 11.4726 | 24.2367 | 40.1288 | 58.2890 |
| | Touratier [15] | 3.0007 | 11.4735 | 24.2407 | 40.1403 | 58.3144 |
| | Karama [18] | 3.0010 | 11.4771 | 24.2561 | 40.1806 | 58.3957 |
| | Soldatos [17] | 3.0006 | 11.4726 | 24.2364 | 40.1279 | 58.2869 |
| 30 | Ambartsumian [12] | 3.0454 | 12.1132 | 27.0062 | 47.4207 | 72.9772 |
| | Kruszewski [13] | 3.0454 | 12.1132 | 27.0062 | 47.4207 | 72.9772 |
| | Reddy [14] | 3.0454 | 12.1132 | 27.0062 | 47.4207 | 72.9772 |
| | Touratier [15] | 3.0454 | 12.1133 | 27.0067 | 47.4222 | 72.9810 |
| | Karama [18] | 3.0454 | 12.1138 | 27.0090 | 47.4293 | 72.9974 |
| | Soldatos [17] | 3.0454 | 12.1132 | 27.0062 | 47.4206 | 72.9771 |

**Table 7.** Comparison of non-dimensional frequency of thickness shear mode $\overline{\omega}_\varphi$ of the piezoelectric beam for various modes of vibration.

| Aspect Ratio S | Model | Modes of Vibration | | | | |
|---|---|---|---|---|---|---|
| | | *m*=1 | *m*=2 | *m*=3 | *m*=4 | *m*=5 |
| 2 | Ambartsumian [12] | 9.8072 | 14.9405 | 20.6814 | 26.6380 | 32.6955 |
| | Kruszewski [13] | 9.8072 | 14.9405 | 20.6814 | 26.6380 | 32.6955 |
| | Reddy [14] | 9.8072 | 14.9405 | 20.6814 | 26.6380 | 32.6955 |
| | Touratier [15] | 9.8022 | 14.9352 | 20.6761 | 26.6328 | 32.6905 |
| | Karama [18] | 9.8048 | 14.9352 | 20.6746 | 26.6305 | 32.6877 |
| | Soldatos [17] | 9.8080 | 14.9411 | 20.6819 | 26.6385 | 32.6959 |
| 5 | Ambartsumian [12] | 47.5546 | 55.9525 | 67.1314 | 79.8167 | 93.3781 |
| | Kruszewski [13] | 47.5546 | 55.9525 | 67.1314 | 79.8167 | 93.3781 |
| | Reddy [14] | 47.5546 | 55.9525 | 67.1314 | 79.8167 | 93.3781 |
| | Touratier [15] | 47.5255 | 55.9222 | 67.1000 | 79.7845 | 93.3454 |
| | Karama [18] | 47.5546 | 55.9423 | 67.1118 | 79.7896 | 93.3452 |
| | Soldatos [17] | 47.5599 | 55.9575 | 67.1361 | 79.8210 | 93.3820 |
| 10 | Ambartsumian [12] | 180.3669 | 190.2187 | 205.1441 | 223.8103 | 245.1810 |
| | Kruszewski [13] | 180.3669 | 190.2187 | 205.1441 | 223.8103 | 245.1810 |
| | Reddy [14] | 180.3669 | 190.2187 | 205.1441 | 223.8103 | 245.1810 |
| | Touratier [15] | 180.2522 | 190.1022 | 205.0251 | 223.6889 | 245.0572 |
| | Karama [18] | 180.3821 | 190.2186 | 205.1239 | 223.7694 | 245.1204 |
| | Soldatos [17] | 180.3882 | 190.2396 | 205.1645 | 223.8301 | 245.2002 |
| 30 | Ambartsumian [12] | 1595.5449 | 1606.0432 | 1623.3029 | 1646.9948 | 1676.7049 |
| | Kruszewski [13] | 1595.5449 | 1606.0432 | 1623.3029 | 1646.9948 | 1676.7049 |
| | Reddy [14] | 1595.5449 | 1606.0432 | 1623.3029 | 1646.9948 | 1676.7049 |
| | Touratier [15] | 1594.5170 | 1605.0133 | 1622.2698 | 1645.9573 | 1675.6621 |
| | Karama [18] | 1595.7288 | 1606.2091 | 1623.4397 | 1647.0929 | 1676.7564 |
| | Soldatos [17] | 1595.7368 | 1606.2347 | 1623.4939 | 1647.1849 | 1676.8941 |

From the above results, it follows that the values of flexural frequencies $\omega_w$ and thickness shear mode frequencies $\omega_\varphi$ are in excellent agreement with each other for all modes of vibration.

## 7. Potential Applications and Future Work

The unified shear deformation theory can contribute to a more accurate design of composite piezoelectric sensors and actuators for various mechanical applications.

The work will be extended at first to the bending and vibration problems of composite piezoelectric beams and plates with different important boundary conditions (clamped–clamped and clamped–free) and thermal effects will also be included.

Another extension of substantial interest will involve nonolocal strain gradient theory for the study of functionally gradient materials in micro- and nanoscale structures with size effects [32,33].

## 8. Conclusions

In this work, six models of elastic beam with high order shear deformation terms were extended for piezoelectric materials and grouped. The unified theory for shear deformation and electric potential distribution was used for the analysis of the static bending problem and the free flexural vibration problem of a piezoelectric beam with simply supported boundary conditions. From the study and comparison of the related numerical results, the following conclusions are drawn:

1.  The transverse displacement $w$ is maximum at the middle points of the beam ($x = L/2$, $z$) and the electric potential $\widetilde{\varphi}$ is maximum in the middle plane of the beam ($x$, $z = 0$), for all models and aspect ratios $S$.
2.  The general model underestimates the maximum transverse displacement $w$ for very thick beams ($S = 2$) and overestimates the $w$ for thick ($S = 5$, $S = 10$) and slender ($S = 30$) beams. The maximum electric potential $\widetilde{\varphi}$ predicted by the general model is in very good agreement with the exact solution for slender beams ($S = 30$).
3.  The maximum transverse shear stress $\tau_{xz}$ predicted by the Touratier model is in very good agreement with the corresponding result from the exact solution. The models of Ambartsumian, Reddy and Soldatos underestimate while the model of Karama overestimates the value of transverse shear stress for all aspect ratios.
4.  The results of flexural frequencies $\omega_w$ and thickness shear mode frequencies $\omega_\varphi$ are in excellent agreement with each other for all modes of vibration.

**Author Contributions:** Conceptualization and methodology, K.I.N. and E.P.H.; investigation and analysis, K.I.N., K.G.B. and E.P.H.; software and writing—original draft preparation, K.I.N. and E.P.H.; supervision, writing—review and editing, K.G.B. and E.P.H. All authors have read and agreed to the published version of the manuscript.

**Funding:** The implementation of this research is co-financed by Greece and the European Union (European Social Fund-ESF) through the Operational Programme «Human Resources Development, Education and Lifelong Learning» in the context of the Act "Enhancing Human Resources Research Potential by undertaking Doctoral Research" Sub-action 2: IKY Scholarship Programme for PhD candidates in Greek Universities (MIS-5113934).

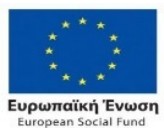

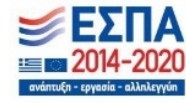

**Data Availability Statement:** Data are available from the corresponding authors upon reasonable request.

**Conflicts of Interest:** The authors declare no conflict of interest. The funders had no role in the design of the study; in the collection, analyses or interpretation of the data; in the writing of the manuscript; or in the decision to publish the results.

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
