# Peer review of "Static and Dynamic Analysis of Linear Piezoelectric Structures Using Higher Order Shear Deformation Theories"

_jcs, doi:10.3390/jcs7020087_

Round 1

Reviewer 1 Report

This manuscript is clearly presented, but this topic is traditional and lacks novelty, the authors didn't present the main contributions of this work in the manuscript. In addition, the resolution of the figures in the result section is low and not clear to the readers. Therefore, I don't recommend its publication.

Author Response

Please see the attached file with our responses to the comments of the Reviewer 1

Yours sincerely,

Evangelos P. Hadjigeorgiou, PhD and

Konstantinos G. Beltsios, PhD

corresponding authors

Reviewer 2 Report

The manuscript presents the static and dynamic analysis of linear piezoelectric structures based on a unified displacement field for higher-order shear deformation theories. A general mathematical model is derived. Although the structure and presentation of the manuscript are clear, following comments/issues must be carefully addressed in the revision before it can be considered for publication in JCS:

(1) The presentation in the Introduction Section to pinpoint the research gap and the new contribution of the present work is somewhat unclear. The authors are recommended to revise this section to clearly address deficiencies or shortcomings of existing studies and also highlight the novelty of their work. The new research findings that are of practical importance in engineering applications and advance the knowledge base in the subject area should be pointed out.

(2) Please sufficiently motivate the choice of your formulation.

(3) The presented work is a comparison study, and I could not find the aim of this study. More elaboration on this issue is required.

(4) Please check the correctness of Eq. (4) and Eq. (11).

(5) Most of the equations were written directly without references. Please provide proper citations to all equations taken directly from the literature. In addition, all variables and symbols appearing in all equations must be clearly defined once first introduced.

(6) More detailed discussion on obtained results reported in Section 6 is required.

(7) Some conclusions are quite obvious (e.g., conclusions 2, 3, and 4) and should be removed. It is recommended to highlight only the major findings.

Author Response

Please see the attached file with our detailed responses to the comments of the Reviewer 2

Yours sincerely,

Evangelos P. Hadjigeorgiou, PhD,

and Konstantinos G. Beltsios, PhD,

corresponding authors

Reviewer 3 Report

I have attached file to this email

Author Response

Please see the attached file with our detailed responses to the comments of the Reviewer 3

Yours sincerely,

Evangelos P. Hadjigeorgiou, PhD,

and Konstantinos G. Beltsios, PhD,

corresponding authors

Reviewer 4 Report

The authors have done great work. 

It needs some improvement to publish the paper.

Other than else, nothing. I will recommend this paper for publications. 

Please check the references and equations once. 

Author Response

Please see the attached file with our responses to the comments of the Reviewer 4

Yours sincerely,

Evangelos P. Hadjigeorgiou, PhD,

and Konstantinos G. Beltsios, PhD,

corresponding authors

Round 2

Reviewer 1 Report

Since the main contributions of this manuscript are not satisfactory, I insist on my first decision.

Reviewer 3 Report

The corrections are done

Reviewer 4 Report

Authors have done all the corrections.